# Strength Characteristics and Microstructure Analysis of Alkali-Activated Slag–Fly Ash Cementitious Material

**DOI:** 10.3390/ma15176169

**Published:** 2022-09-05

**Authors:** Chenhui Zhu, Yuanyuan Wan, Lei Wang, Yuchen Ye, Houjun Yu, Jie Yang

**Affiliations:** 1School of Transportation and Civil Engineering, Nantong University, Nantong 226019, China; 2Nantong Construction Engineering Quality Testing Center, Nantong 226019, China; 3Suyu District Water Conservancy Bureau of Suqian City, Shuyang 223800, China

**Keywords:** alkali activation, cementitious materials, strength characteristics, microstructure

## Abstract

Modifying the admixture of alkali-activated cementitious materials using components such as fly ash and fine sand may reduce CO_2_ emissions and conserve natural resources and energy. This study adopted strength testing, scanning electron microscopy, and mercury intrusion porosimetry to investigate the influence of different admixtures on the compressive strength and flexural strength of alkali slag cementing materials and the microstructure characteristics of hardened slurry under the action of load. The flexural strength of alkali slag cement slurry and mortar was reduced by replacing slag powder with fly ash. Content of fine sand less than 20% had little effect on the strength of alkali slag cement mortar; however, when the content of fine sand exceeded 30%, the strength decreased significantly. The hydration degree at 3 d was large, and the density of slurry increased with the extension of age. Increased fly ash or fine sand content decreased the density of the slurry, and increased fly ash resulted in a large number of unhydrated fly ash particles in the cementitious materials. Addition of fine sand resulted in a large number of microcracks in the slurry, which gradually decreased with the extension of hydration age.

## 1. Introduction

The uncontrolled emission of CO_2_ will sharply deteriorate the living environment of human beings, contrary to the scientific concept of development of “people-oriented, comprehensive, coordinated and sustainable development.” [1] Use of traditional Portland cement concrete in modern industry and agriculture, transportation, urban infrastructure construction, hydropower engineering, and the comprehensive development of marine resources produces large amounts of CO_2_, with emissions exceeded only by the coal and steel industries [2]. Continued production and use of traditional Portland cement concrete will increase the Earth’s greenhouse effect, causing serious pollution to the environment.

Chemical-trigger off-gelled materials are formed using a chemical agent (generally an alkaline substance, leading to the common name alkali-activated gelled materials) added to all kinds of silicate and aluminum silicate minerals, as well as industrial slag powder. By adding the right amount of water and sand, stone, or fiber material, these materials become a plastic slurry after mixing, and can condense in air or water into a solid gel material form that has certain mechanical strength. These materials have many advantages, such as high early strength, fire resistance, corrosion resistance, impermeability and frost resistance, and strong bonding with old concrete [3,4,5,6,7]. Adding alkali-activated cementitious materials to concrete can reduce natural resource consumption, CO_2_ emissions, and energy consumption [8].

From the 1950s, scholars began to study the development and application of alkali cementing materials: By mixing low-temperature calcined kaolin and limestone with an alkaline liquid, the French scientist Davidovits produced zeolite materials with good strength and new aluminosilicate bonds, which he named Geopolymer [9]; Roy studied the fast-hardening alkali slag cementitious material, and analyzed its structure and properties [10]. Wang and Scrivener systematically studied the products and microstructure of slag activated by alkaline substances [11]. In the 1980s, preliminary research on alkali-activated cementing materials began in China: Pu Xincheng et al. conducted research on the durability of alkali-slag cement concrete (including sulfate, seawater erosion, frost resistance, and impermeability) [12,13,14]. The results showed that the alkali slag cement concrete has excellent sulfate resistance, good seawater resistance, and far better impermeability than ordinary Portland cement. Fang Yonghao et al. systematically studied the shrinkage and toughening of alkali phosphorus slag/slag cementitious materials. They showed that low-temperature curing could reduce the shrinkage and cracking of alkali slag cementitious materials. The shrinkage of alkali slag cement mortar can be significantly reduced, and its mechanical properties can be improved, by replacing sand with fine particles of slag or phosphorus slag of 0.08–0.63 mm [15,16,17,18]. Chen Yiqun [19,20] proposed adding fly ash into alkali slag cementing material to reduce the chemical shrinkage of slurry. The shrinkage reducing agent and magnesium oxide expansion agent reduce the drying shrinkage of alkali slag cementing material, and the magnesium oxide expansion agent with appropriate dosage presents a compensation shrinkage effect in the later stage of slurry curing. In recent years, the research on alkali-activated cementitious materials mainly focuses on two directions. First, continue to study alkali-activated slag cementitious materials to improve their properties and expand their applications [21,22,23]; Second, other solid wastes with alkali-activated cementitious properties are used to prepare new alkali-activated cementitious materials, such as silicomanganese slag, copper and nickel slag, red mud-iron tailings, copper slag, nickel slag, and ferronickel slag [24,25,26,27,28,29,30].

From these research studies and application, the main obstacle to the wide application of alkali-activated cementation materials and concrete is their large shrinkage deformation, creep deformation, and cracking. Therefore, fly ash and fine sand with lower alkali excitation activity than slag powder are selected as admixtures in this paper. This study aimed to analyze the influence of adding different admixtures and adjusting the dosage of admixtures on the compressive and flexural strength of alkali slag cementing material by means of modern testing methods such as strength testing, and scanning electron microscopy (SEM), and to explore the microstructure of the hardened slurry of alkali slag cementing material under the action of load.

## 2. Materials and Methods

### 2.1. Test Scheme and Mix Design

To study the effect of slag powder, fly ash, and fine sand at different dosages in alkali slag cementing materials on compressive strength, flexural strength, and microstructure, three series of experiments were designed. For the determination of the strength of the fly ash cement slurry, series A experiments were conducted. To explore the alkali slag cement–fly ash mortar strength development, series B of experiments were conducted. The effect of different amount of fine sand on alkali slag cement mortar strength was investigated by series C of experiments. The detailed mix ratio is shown in Table 1. The water–cement ratio of the alkali slag cement slurry was 0.3.

### 2.2. Raw Materials

#### 2.2.1. Powdered Ore

The slag powder used in this test was provided by Hunan Huaxin Xiang Steel cement Co., Ltd. The color was gray, slightly lighter than the gray of pulverized coal. The chemical constituents measured by X-ray fluorescence analysis are shown in Table 2. After determination using a 0.08 mm square hole sieve, the sieve allowance was 2.6% and Brewer specific surface area was 455 m^2^/kg.

#### 2.2.2. Fly Ash

The fly ash used in this test was ⅱ fly ash from Lianyuan Power Plant of Huarun. The appearance of the fly ash was gray powder, and the specific surface area was 370 m^2^/kg. The chemical composition of the fly ash measured by X-ray fluorescence analysis is shown in Table 3. Figure 1 shows SEM image of the fly ash.

#### 2.2.3. Fine Sand

In this study, Nanjing Liuhe river sand was replaced by fine sand. Through the screening and grading of sand, the particle size range of the fine sand was 0.08–0.63 mm.

#### 2.2.4. Water Glass

The water glass (WG) used in this test was the product of Nanjing Chemical Plant, with modulus 3.28 and solid content 27.43%. See Table 4 for its chemical composition. The WG was diluted with water and then NaOH (analytical pure AR grade produced by Shanghai Shiyi Chemical Reagent Co., Ltd.) was added to make dilute WG with a modulus of 1.40. The *n* in Na_2_O·*n*SiO_2_ is the molar ratio of silicon dioxide and alkali metal oxides, that is, the modulus.

#### 2.2.5. Standard Sand

The sand used in the strength tests of alkali slag cementing material was Chinese ISO standard sand. The sand used in drying shrinkage and autogenous shrinkage tests of alkali slag cementing material was fine standard sand with a particle size range of 0.5–1.0 mm in accordance with GB/17671-1999.

#### 2.2.6. Mixing Water

The mixing water used in testing was the tap water of the laboratory.

### 2.3. Test Method

#### 2.3.1. Preparation of Specimens

Because of the high temperature sensitivity of alkali slag cement, all raw materials were placed in a curing box at 20 °C for 6 h in advance before the preparation of the specimen. After the raw materials reached the preset temperature, they were taken out quickly and molded. After demudding, they were maintained to the specified age according to the test method.

#### 2.3.2. Test Method for Mechanical Properties

The alkali slag cementing material mortar strength test was performed according to GB/T17671-1999 “Cement Mortar Strength Test Method (ISO method)” using a 40 mm × 40 mm × 160 mm prismatic specimen. The slurry strength test used a 20 mm × 20 mm × 80 mm specimen size. The dosage of sodium silicate to the activator with modulus 1.4 was calculated by Na_2_O equivalent, accounting for 6% of the mass of the active cementing material (slag powder + fly ash). The molding water amount was added according to the water–glue ratio, deducting the water amount brought in by sodium silicate, and the flow degree of mortar was controlled as 180–190 mm. The specimen was cured in a standard curing tank for 24 h with a mold, and after demolding was placed in 5% NaOH solution in a constant-temperature (20 ± 1 °C) water bath tank for curing until the specified age of 3, 28, or 90 d. After reaching the curing age, it was removed and wiped on the surface of the specimen, and then covered with a wet cloth until the test. The samples of compressive strength and flexural strength were the same. The samples were first tested for flexural strength according to the standard (a group of three samples), and the broken samples were used for the compressive strength test (a group of six samples).

#### 2.3.3. Microstructural Analysis Methods

Alkali slag cement slurry and mortar samples were prepared according to the ratios specified in Table 1, and cured to the specified age in the standard curing chamber. The uncarbonized part inside the sample was soaked in anhydrous ethanol solution for 24 h to terminate its hydration. For SEM analysis, the relatively flat particle samples with non-formed surfaces were selected and vacuumized in a vacuum drying oven at 60 °C for 6 h, and observed after spraying gold.

## 3. Results and Discussion

### 3.1. Mechanical Properties of Alkali Slag Cementitious Materials

Under the condition of the same fluidity, the water requirement of the ore powder–fly ash and ore powder–fine sand mixed cementitious systems decreased with increasing fly ash or fine sand content. The main reason for this phenomenon is that the specific surface area of fly ash and fine sand is much smaller than that of ore powder, and the water requirement of slurry of the corresponding system is therefore much smaller. Moreover, the reactivity of fly ash is not as good as that of slag powder, and the fine sand has almost no activity. In the early reaction phase, the hydration of slag powder is the main reaction. When the content of fly ash and fine sand increases, the content of slag powder in the corresponding cementitious system inevitably decreases, and the content of cementitious materials involved in the early hydration reaction consequently also decreases, decreasing the water requirement under the condition of the same fluidity. In addition, the spherical particles of fly ash have a “ball effect,” which can reduce the friction between particles in the mixed system, play a role of dispersion and lubrication, and improve the flow performance of alkali slag cement mortar. This phenomenon also causes the water requirement of slurry to decrease with increasing fly ash content. The decreased water requirement is beneficial to increasing the strength.

#### 3.1.1. Effect of Fly Ash Content on Strength of Alkali Slag Cement Paste

Series A tests were performed, as shown in Table 1. After the molded samples reached the corresponding age, they were taken out of the curing box, and the compressive strength and flexural strength were tested after the specimen surface was wiped free of water droplets. The test results were averaged after data outside the error range were removed, and the results are shown in Figure 2.

It can be seen from Figure 2 that the compressive strength of alkali slag cement slurry at different ages decreased with increasing fly ash content; in particular, the early strength decreased greatly. In general, when the fly ash content was less than 50%, it had little influence on the compressive and flexural strength of the cementing material. When the fly ash content was 50%, the compressive strength at 3, 28, and 90 d was 50, 80, and 102 MPa, respectively. When the fly ash content exceeded 50%, the compressive and flexural strength decreased rapidly. When fly ash content was 80%, the 3, 28, and 90 d compressive strength were 71.4%, 49.4%, and 39.0% lower than that of the gelled system with pure ore powder, respectively. The effect on flexural strength was similar. The compressive and flexural strength of the same ratio of alkali slag cement paste increased with the extension of curing time, and no strength shrinkage phenomenon was observed in the later period.

The test results show that the strength of alkali slag cement slurry decreases with substitution of fly ash, and the higher the fly ash content, the lower the strength of the alkali slag cement slurry. In the early stage of low hydration degree, the strength of alkali slag cement slurry with fly ash substitute decreases greatly. 

#### 3.1.2. Effect of Fly Ash Content on Strength of Alkali Slag Cement Mortar

Series B tests were carried out according to Table 1. The influence of different fly ash content on the compressive and flexural strength of alkali slag cement mortar at 3, 28, and 90 d is shown in Figure 3.

As shown in Figure 3a,b, when fly ash replaced ore powder, the compressive and flexural strength of mortar at different ages decreased with the increase of fly ash content, especially in the early stage. In general, when the fly ash content was less than 50%, it had little effect on the compressive and flexural strength of the cementing material. When the fly ash content was 50%, the compressive strength at 3, 28, and 90 d was 36, 73, and 84 MPa, respectively. When the fly ash content exceeded 50%, the compressive and flexural strength decreased rapidly. When the fly ash content was 80%, the compressive strength at 3, 28, and 90 d was 15, 35, and 58 MPa, respectively. The influence of fly ash content on the flexural strength of mortar also followed a similar rule. The compressive and flexural strength of the same ratio of alkali slag cement mortar increased with the extension of curing time, and no strength shrinkage phenomenon was observed in the later period.

It can be seen that the replacement of slag powder by fly ash causes a decline in the strength of the alkali slag cement mortar, and the higher the fly ash content, the lower the strength of the alkali slag cement mortar. Especially in the early stage of low hydration degree, the strength of the alkali slag cement mortar decreased greatly. When the fly ash content was less than 50%, it had little influence on the strength of the cementing materials. When the fly ash content exceeded 50%, the compressive and bending strength of specimens decreased significantly. Considering practical application, economic benefits, and energy conservation, these findings suggest that the fly ash content should be about 50%.

#### 3.1.3. Effect of Sand Content on Strength of Alkali Slag Cement Mortar

Series C tests were carried out according to Table 1. Figure 4 shows the compressive and flexural test results of alkali slag cement mortar with 0.08–0.63 mm fine sand replacing ore powder at different ages.

It can be seen from Figure 4a that when the content of fine sand was 5–20%, the compressive strength of the specimen at hydration age of 3 d was between 59 and 62 MPa, which is very similar to the compressive strength of the alkali slag cement mortar specimen without fine sand (62 MPa). The compressive strength values of specimens at hydration ages of 28 d and 90 d also showed a similar pattern. When the content of fine sand reached more than 30%, the compressive strength curve exhibited a steep downward trend. This phenomenon shows that when the fine sand content was not more than 20%, the effect of increased fine sand content on the compressive strength of the alkali slag cement was not large. The sclerotium basic cemented together by the hydration reaction to generate gel products. However, when the dosage was 30% or more, the reduced dosage of slag powder resulted in a gelled material reduction; hydration-generated gel product was reduced. More pore pits were formed between fine sand particles, and the gel products could not cross-lap to form a network structure, resulting in a low density of slurry and decreased strength. The bending strength curve in Figure 4b has a trend similar to the compressive strength. When the content of fine sand exceeded 30%, the bending strength decreased more, suggesting that the optimal content of fine sand should be controlled at about 20%.

### 3.2. Analysis of Microscopic Results

Figure 5 shows SEM images of the alkali slag cement slurry with pure ore powder at the hydration age of 3 d. As can be seen from Figure 5a, the structure of the slurry of the alkali slag cementing material was already very dense after hydration for 3 d, and the cement stone was mainly composed of dense gel and unhydrated slag particles. From the results reported in Section 3.1, the compressive strength of 3 d alkali slag cement slurry reached 75 MPa, indicating that the structure of cement stone was very dense, which is consistent with the structure analyzed by SEM.

Figure 6 shows the SEM images of the alkali slag cement slurry with 50% fly ash at hydration age of 3 d. Figure 6a,b show that the structure of the cement stone was dense, but a large number of fly ash particles were piled together. The fly ash particles were spherical with a smooth surface and regular, and there was no dissolution phenomenon. This indicates that the activity of fly ash is not as good as that of slag powder, and the hydration reaction of slag powder stimulated by sodium silicate takes place first, while the pozzolanic reaction of fly ash hardly takes place at this stage. In the slag powder–fly ash mixed cementitious system, there are a lot of fly ash particles not entirely in the hardened cement paste and no reaction of fly ash particles. Moreover, the system has not yet formed gel product including fly ash particles, and the porosity is high with many pore pits, resulting in low compactness. This is consistent with the corresponding 3 d age compressive strength of only 50 MPa, which is lower than the strength of the pure ore powder system at the same age.

Figure 7 shows the SEM images of 80% fly ash alkali slag cement slurry at the hydration age of 3 d. Compared with Figure 6a, due to the increased fly ash content, more unreacted fly ash particles were accumulated in Figure 7a, and a large number of microcracks, pores, and pits existed in the hardened slurry of the system. The compactness of the whole slurry was lower, and the corresponding strength was also lower, which is consistent with the corresponding compressive strength of 21 MPa at 3 d hydration age. It can also be observed from Figure 7a that unhydrated fly ash particles fill the voids in the hardened slurry and play a micro-aggregate effect in the early stage of hydration.

Figure 8 and Figure 9 show SEM images of alkali slag cement mortar with hydration ages of 3 d and 28 d, respectively, for a cementitious system containing pure ore powder. Figure 8a indicates that under the condition of sodium silicate alkaline solution, the slag has undergone a certain degree of hydration reaction, resulting in a large amount of amorphous gel, and the structure of the hardened slurry is relatively dense. Comparing Figure 8 and Figure 9a reveals that the structure of hardened cement paste shown in Figure 8a is significantly denser than that in Figure 8a; good bonding existed between the cement and aggregate, the aggregate was closely included in the hydration products, the structure of natural gas hydrates was compact, and the hydrates mutually intertwined with the matrix forming a network. In macroscopic performance, 28 d compressive strength was about 30% higher than 3 d compressive strength.

Figure 10 and Figure 11 show SEM images of alkali slag cement sand substituted with 50% fly ash at hydration ages of 3 d and 28 d, respectively. Figure 10a shows that the structure of hardened slurry was relatively loose, and many unreacted fly ash particles existed in the system. Figure 10b shows that fly ash particles were smooth and regular spheres, and the gel generated by hydration of the slag powder was wrapped on the surface of fly ash particles, while the solubility of Ca(OH)_2_ generated by hydration was too low to damage the vitreous structure of the fly ash, and no corrosion occurred on the surface. This indicates that fly ash hardly exhibited the pozzolanic effect in the early hydration stage, but filled in the slurry as a fine aggregate. Due to the short hydration age, not enough gel products have been formed in the system to wrap and cement fly ash particles. The cementation between hydration products was not sufficient, and there were pores and pits in the system, so the density of the slurry was very low and the strength was correspondingly low.

However, with the extension of hydration age, hydration gel products in the system increased and cross-lapped with each other, porosity was significantly reduced, and the structure became increasingly dense. Fly ash was depolymerized because the slag is excited by sodium silicate, and the dissolved Ca^2+^ and Mg^2+^ ions react with fly ash vitreous in a pozzolanic manner. It can be seen from Figure 11a that the fly ash particles that had not undergone a hydration reaction in the hardened slurry decreased significantly after 28 d, but still existed. Figure 11b shows that dissolution occurred on the surface of the fly ash particles by this hydration age.

Figure 12 and Figure 13 show SEM images of alkali slag cement sand replaced by 80% ore powder with fly ash at hydration ages of 3 d and 28 d, respectively. As can be seen from Figure 12a, the structure of the hardened slurry was relatively loose, and flocculent gel products covered the surface of the slag and fly ash particles, but there was no dissolution phenomenon observed on the fly ash surface. A large number of unreacted fly ash particles were stacked together as smooth spherules and filled in the gaps in the hardened slurry, accompanied by pores, pits, and cracks in the system. With the extension of hydration age, it can be seen from Figure 13a that a large number of gels were generated in the hardened slurry, and the gels overlapped and crossed each other, indicating that they continuously developed and accumulated, filling holes, pores, and cracks, and cementing the unhydrated particles together. The unhydrated fly ash particles were also significantly reduced. The 28 d compressive strength of 80% fly ash–substituted alkali slag cement mortar was 20 MPa higher than its 3 d strength, which is consistent with the SEM image analysis results. At the same hydration age, increased fly ash content decreased the densification degree of the hardened slurry structure, and the mortar sample with fly ash content of 80% had the lowest density, with many pores and pits. In terms of macroscopic properties, the 3 d compressive strength was 15 MPa, which is 47 MPa and 21 MPa lower than that of alkali slag cement mortar with pure ore powder and 50% ore powder substituted by fly ash at the same age. The 28 d compressive strength was 35 MPa, which is 46 MPa and 38 MPa lower than the same comparison specimens, respectively.

### 3.3. Discussion

The results consistently indicate that the cementing material of alkali slag with hydration age of 3 d has been hydrated to a great extent, and the structure of the early hardened slurry was already very dense. The density of the slurry increased with the extension of hydration age. With increasing fly ash content, the density of the slurry decreased, and a large number of unhydrated fly ash particles appeared in the SEM images of 3 d hydration age, and these particles still existed until 28 d.

From the point of view of hydration mechanism, slag with fly ash has potential hydraulic properties and contains more CaO. In the process of quenching, fly ash is the network regulator in the structure of slag glass when slag vitreous is formed. Increased fly ash content reduces the degree of polymerization of network ions, which is beneficial to slag activity. At the initial stage of hydration, Ca, Mg, and Al in the silica-poor phase of slag dissolve into active cations owing to the hydrolysis of sodium silicate as the activator. With increased slurry pH, the silica-rich phase of slag becomes hydrated, and these active cations and Si ions form gel hydration products. At the same time, [SiO_4_]^4–^ ions generated by hydrolyzation of sodium silicate can solve the problem of low hydration products and low slurry strength at the initial stage of hydration. On the other hand, the fly ash used in this test has low CaO content. Owing to the small CaO/SiO_2_ ratio, the polymerization degree of [SiO_4_]^4–^ ions in the vitreous is high, forming a relatively continuous three-dimensional network structure in which and Al^3+^ also participates. The total amount of oligomer [SiO_4_]^4–^ ions is too small. It is generally believed that this is the main reason why the activity of fly ash is lower than that of slag. Therefore, with the incorporation of fly ash, the amount of hydration products in the early hardened slurry decreases because fly ash does not participate in the reaction at the early hydration stage. In addition, the chemical reaction of the active components in fly ash particles is slow, the water film between particles has not been filled, the hydration products have not been connected, and more voids and open pores make the compactness of the slurry poor, so the early strength of alkali slag–fly ash cement slurry is lower than that of alkali slag cement slurry. With the extension of time and the continuous hydration reaction, the slag powder is stimulated and depolymerized by sodium silicate, and the dissolved Ca^2+^ and Mg^2+^ ions react with the fly ash vitreous in pozzolanic ash, wrapping the unreacted substances into the structure to obtain high-strength hydration products, which can supplement and provide long-term strength. Generally speaking, the ash pozzolanic effect is gradually reflected after 7 d [31,32].

## 4. Conclusions

This study investigated the influence of different admixtures on the compressive and flexural strength of alkali slag cement slurry and mortar. According to the test results, the following conclusions can be drawn:(1)The substitution of fly ash for ore powder progressively reduces the compressive and flexural strength of alkali slag cement slurry and mortar. When the fly ash content was less than 50%, it had little influence on the strength of the cementing materials. When the fly ash content exceeded 50%, the compressive and bending strength of the specimens decreased significantly. Considering the practical application, economic benefits, and energy conservation, it is suggested that the fly ash content should be approximately 50%.(2)Fine sand may be used to replace ore powder. When the content of fine sand was less than 20%, the strength of the alkali slag cement mortar changed little. When the content of fine sand exceeded 30%, the compressive and flexural strength of the alkali slag cement mortar decreased significantly. The alkali slag cement specimens with different proportions of fine sand all had higher early strength. With the extension of curing age, the alkali slag cement specimens exhibited no late strength shrinkage phenomenon.(3)SEM analysis showed that the cementing material of alkali slag with hydration age of 3 d has been hydrated to a great extent, and the structure of the early hardened slurry was already very dense. The density of the slurry increased with the extension of hydration age.(4)With increasing fly ash content, the density of the slurry decreased, and a large number of unhydrated fly ash particles appeared in the SEM images of 3 d hydration age, and these particles still existed until 28 d. With increasing fine sand content, the density of the slurry decreased, and a large number of microcracks with a width of about 1 μm appeared in the SEM image of 3 d hydration age. With the extension of hydration age, the microcracks gradually decreased.

Next, this topic will study the influence of fly ash and fine sand content on the early shrinkage performance of alkali-activated slag cementitious material.

## Figures and Tables

**Figure 1 materials-15-06169-f001:**
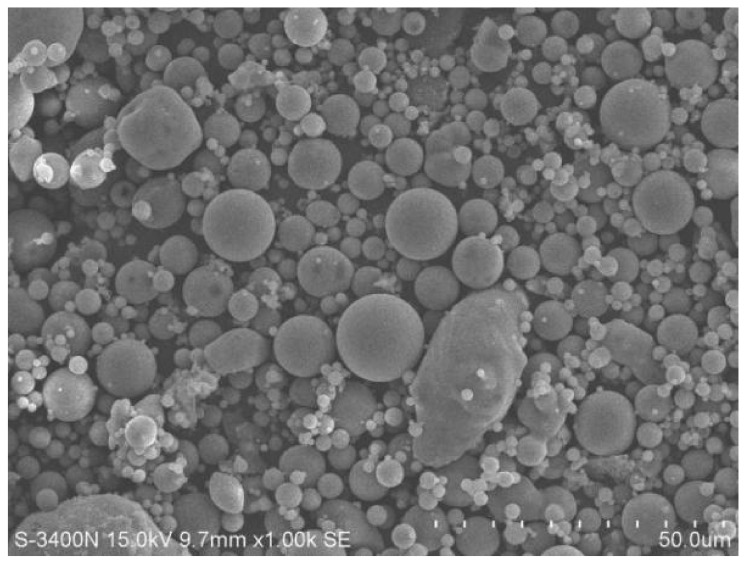
SEM image of fly ash.

**Figure 2 materials-15-06169-f002:**
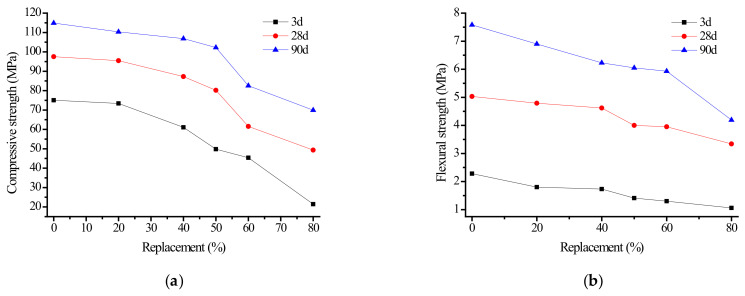
Effect of fly ash content on strength of alkali slag cement paste. (**a**) Compressive strength, (**b**) Flexural Strength.

**Figure 3 materials-15-06169-f003:**
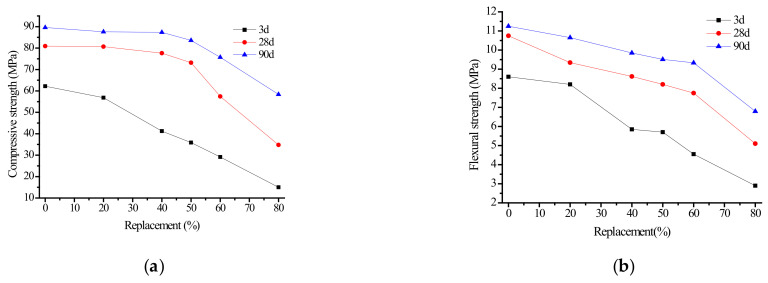
Effect of fly ash content on strength of alkali slag cement mortar. (**a**) Compressive strength, (**b**) Flexural Strength.

**Figure 4 materials-15-06169-f004:**
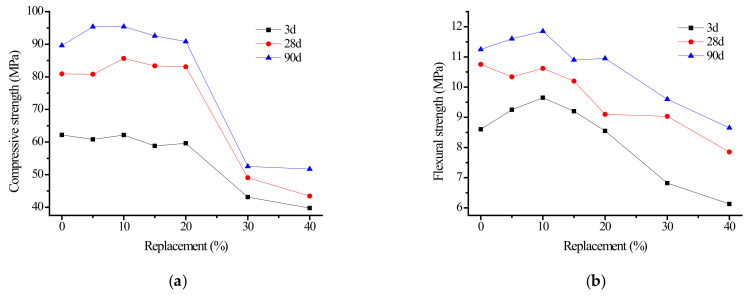
Effect of sand content on strength of alkali slag cement mortar. (**a**) Compressive strength, (**b**) Flexural Strength.

**Figure 5 materials-15-06169-f005:**
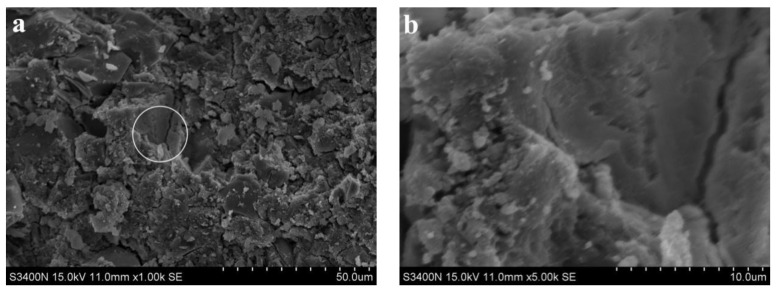
SEM images of the alkali slag cement slurry with pure ore powder at the hydration age of 3 d. (**a**) ×1.00 k, (**b**) ×5.00 k.

**Figure 6 materials-15-06169-f006:**
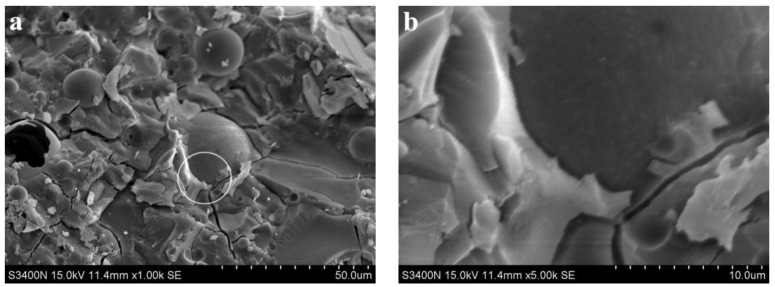
SEM images of the alkali slag cement slurry with 50% fly ash at hydration age of 3 d. (**a**) ×1.00 k, (**b**) ×5.00 k.

**Figure 7 materials-15-06169-f007:**
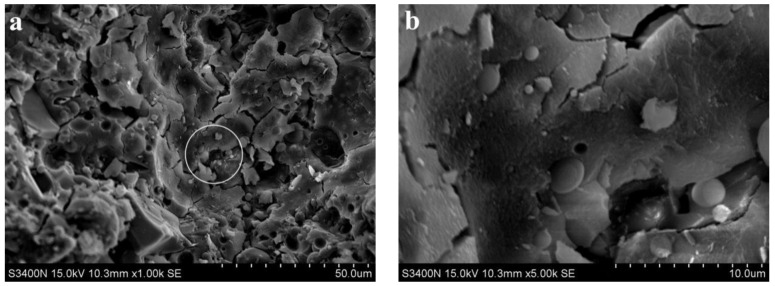
SEM images of 80% fly ash alkali slag cement slurry at the hydration age of 3 d. (**a**) ×1.00 k, (**b**) ×5.00 k.

**Figure 8 materials-15-06169-f008:**
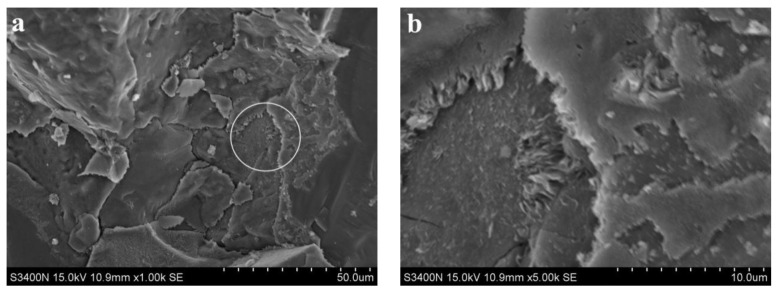
SEM images of alkali slag cement mortar with hydration ages of 3 d. (**a**) ×1.00 k, (**b**) ×5.00 k.

**Figure 9 materials-15-06169-f009:**
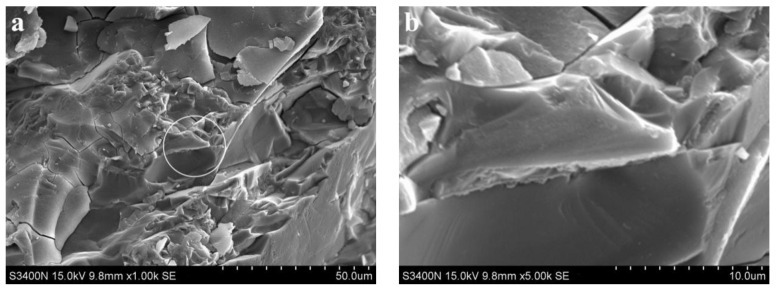
SEM images of alkali slag cement mortar with hydration ages of 28 d. (**a**) ×1.00 k, (**b**) ×5.00 k.

**Figure 10 materials-15-06169-f010:**
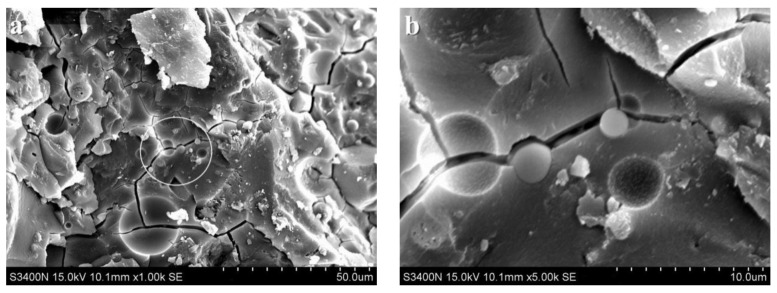
SEM images of alkali slag cement sand substituted with 50% fly ash at hydration ages of 3 d. (**a**) ×1.00 k, (**b**) ×5.00 k.

**Figure 11 materials-15-06169-f011:**
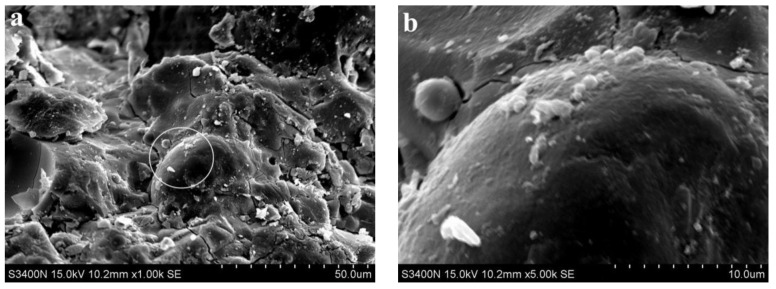
SEM images of alkali slag cement sand substituted with 50% fly ash at hydration ages of 28 d. (**a**) ×1.00 k, (**b**) ×5.00 k.

**Figure 12 materials-15-06169-f012:**
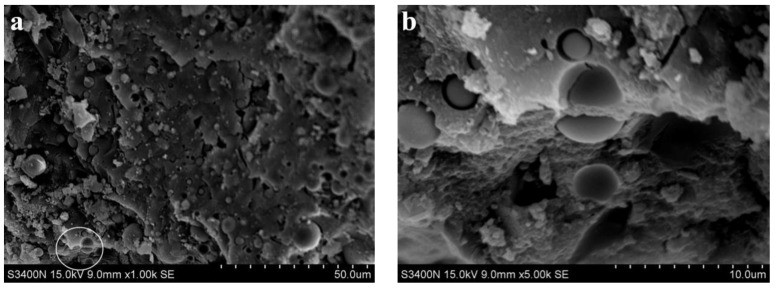
SEM images of alkali slag cement sand replaced by 80% ore powder with fly ash at hydration ages of 3 d. (**a**) ×1.00 k, (**b**) ×5.00 k.

**Figure 13 materials-15-06169-f013:**
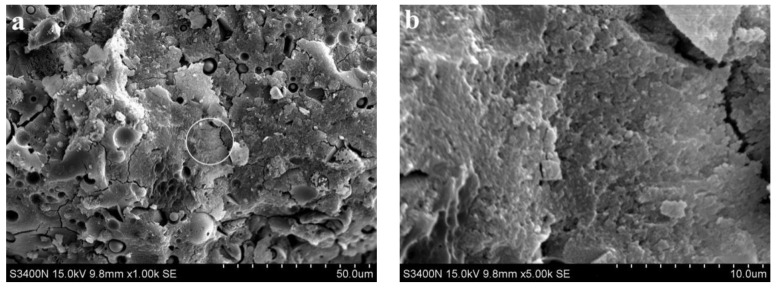
SEM images of alkali slag cement sand replaced by 80% ore powder with fly ash at hydration ages of 28 d. (**a**) ×1.00 k, (**b**) ×5.00 k.

**Table 1 materials-15-06169-t001:** Mix ratio of strength experiment (g).

No.	Slag Powder	Fly Ash	Fine Sand	Standard Sand	Water Glass	Water	Total Water
A1	400	0	0	0	136	40	120
A2	320	80	0	0	136	40	120
A3	240	160	0	0	136	40	120
A4	200	200	0	0	136	40	120
A5	160	240	0	0	136	40	120
A6	80	320	0	0	136	40	120
B1	450	0	0	1350	153	135	225
B2	360	90	0	1350	153	131	220
B3	270	180	0	1350	153	126	216
B4	225	225	0	1350	153	124	214
B5	180	270	0	1350	153	122	211
B6	90	360	0	1350	153	117	207
C1	427.5	0	22.5	1350	146	130	216
C2	405	0	45	1350	138	125	206
C3	382.5	0	67.5	1350	130	120	197
C4	360	0	90	1350	123	115	187
C5	315	0	135	1350	107	105	168
C6	270	0	180	1350	92	95	149

**Table 2 materials-15-06169-t002:** Chemical constituents of slag powder (wt%).

SiO_2_	Al_2_O_3_	Fe_2_O_3_	CaO	MgO	SO_3_	MnO	TiO_2_	LOI
33.23	17.76	0.416	37.17	7.53	3.10	0.404	0.992	—

**Table 3 materials-15-06169-t003:** Chemical constituents of fly ash (wt%).

SiO_2_	Al_2_O_3_	Fe_2_O_3_	CaO	MgO	SO_3_	Na_2_O	P_2_O_5_	LOI
51.07	36.24	2.88	1.25	0.744	0.394	0.438	0.125	3.72

**Table 4 materials-15-06169-t004:** Chemical constituents of water glass (wt%).

Solid Content	Na_2_O	SiO_2_
36.07	8.64	27.43

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
