# Peer review of "Strength Characteristics and Microstructure Analysis of Alkali-Activated Slag–Fly Ash Cementitious Material"

_materials, 2022, doi:10.3390/ma15176169_

Round 1
Reviewer 1 Report
I believe that the paper is suitable for publication in Materials but should be seriously revised according to the comments below.
1. The introduction could be improved, especially in specifying examples of existing alkali-activated slag–fly ash cementitious material disadvantages.
2. How were the mixture ratios chosen? Why these proportions?
3. Why different sample sizes were used for mortar and slurry when determining strength?
4. What were the specific conditions for mechanical testing? Which samples were used for compressive strength and which for flexural strength?
5. It is necessary to clarify on what basis the discussion in section 3.1.1 (last paragraph) is based? Are these assumptions of the authors or established facts?
6. It would be nice to show SEM of the additives before using.
7. The Discussion is too short. What are the features of the obtained results and their difference from other works widely presented in the open literature?
Author Response
Dear reviewer,
Thank you for your useful comments and reviewers’ suggestions on the scientific content of our manuscript. We have carefully revised the manuscript accordingly in the revised version again. We appreciate the detailed and useful comments from you again. The detailed corrections are listed below point by point:
1.The introduction could be improved, especially in specifying examples of existing alkali-activated slag–fly ash cementitious material disadvantages.
Response 1: The introduction has been revised. The disadvantages of existing alkali-activated slag–fly ash cementitious materials such as great shrinkage and easy-cracking are introduced.
2.How were the mixture ratios chosen? Why these proportions?
Response 2: These mixture ratios were chosen because the research group had previously carried out relevant studies on alkali-activated slag cementitious materials, in which the optimal proportion of slag had been obtained. This paper belongs to the follow-up research content, mainly to study the impact of fly ash and fine sand partially replacing slag on alkali-activated slag cementitious materials. Because of the relatively high activity of fly ash, the replacement ratios of 20%, 40%, 50%, 60% and 80% were selected. However, the activity of fine sand is low, so relatively low substitution ratios of 5%, 10%, 15%, 20%, 30% and 40% were selected.
3.Why different sample sizes were used for mortar and slurry when determining strength?
Response 3: This is because the molding mold is different, mortar specimen strength test is according to the standard GB/T17671-1999 "cement mortar strength test method (ISO method)", and the mold of the slurry specimen is homemade.
4.What were the specific conditions for mechanical testing? Which samples were used for compressive strength and which for flexural strength?
Response 4: The demolded specimens were placed in water at 20±1℃ for curing. After reaching the curing age, it was removed and wiped on the surface of the specimen, and then covered with a wet cloth until the test. The samples of compressive strength and flexural strength are the same. The samples will first be tested for flexural strength according to the standard (a group of three samples), and the broken samples will be used for the compressive strength test (a group of six samples).
5.It is necessary to clarify on what basis the discussion in section 3.1.1 (last paragraph) is based? Are these assumptions of the authors or established facts?
Response 5: These discussions are based on the results of strength test and microscopic test in this experiment, combined with the research results of the research group in the early stage. I have moved this paragraph to discussion (section3.3).
6.It would be nice to show SEM of the additives before using.
Response 6: Thank you for your comments, the SEM of fly ash has been added in the paper.
7.The Discussion is too short. What are the features of the obtained results and their difference from other works widely presented in the open literature?
Response 7: Thank you for your valuable comments and the discussion has been revised. The purpose of this study is to solve the shortcoming of fast setting and large shrinkage of alkali-activated slag cementitious materials by adding fly ash or fine sand. The results obtained in this paper, on the one hand, clarify the hydration mechanism of alkali-activated slag–fly ash cementitious materials, and on the other hand, lay a good foundation for the subsequent study of shrinkage properties.
At last, special appreciation to you for these good questions and comments, and all of them give deep insight and understanding about the results of our work, and greatly improved the quality of our paper.
Thanks again.
Sincerely yours,
Chen-Hui Zhu
Reviewer 2 Report
· Add more new references.
· Show the novelty of the paper compared to the literature, however there is still much work on this topic.
· Why you choose these materials?
· In the Introduction section, the last paragraph should contain the scientific contribution and scientific hypotheses of your research. Complete, further elaborate the scientific contribution and scientific hypotheses of your research. Be explicit. In addition to the goal of the research (which was written), the novelty in the context of the scientific contribution should be pointed out. Scientific contributions should be written based on the shortcomings of previous research in the literature. In this way, the authors will better emphasize novelty and scientific soundness.
· Analyze and discuss possibilities of practical application.
· Complete the conclusions with the limitations of the proposed methodology. Also write future research.
· Generally, the quality of the writing could be improved.
Author Response
Dear reviewer,
Thank you for your useful comments and reviewers’ suggestions on the scientific content of our manuscript. We have carefully revised the manuscript accordingly in the revised version again. We appreciate the detailed and useful comments from you again. The detailed corrections are listed below point by point:
1.Add more new references.
Response 1: Thanks for your comments, more new references have been added to the paper.
2.Show the novelty of the paper compared to the literature, however there is still much work on this topic.
Response 2: As you said, it still has a lot of work to do. The final purpose of this project is to solve the problem of great shrinkage of alkali-activated slag cementitious materials, and the addition of fly ash and fine sand is an attempt in this paper. This paper only elucidated its effect on mechanical properties and hydration mechanism, and the subsequent tests on shrinkage properties will be carried out.
3.Why you choose these materials?
Response 3: Because one of the shortcomings of alkali-activated slag cementitious materials is the large shrinkage and easy cracking, fly ash and fine sand are selected in this paper because their activity is lower than that of slag powder. Whether they can reduce the early shrinkage of alkali-activated slag cementitious materials will be studied in the future.
4.In the Introduction section, the last paragraph should contain the scientific contribution and scientific hypotheses of your research. Complete, further elaborate the scientific contribution and scientific hypotheses of your research. Be explicit. In addition to the goal of the research (which was written), the novelty in the context of the scientific contribution should be pointed out. Scientific contributions should be written based on the shortcomings of previous research in the literature. In this way, the authors will better emphasize novelty and scientific soundness.
Response 4: Thank you for your valuable comments. The introduction has been revised as requested.
5.Analyze and discuss possibilities of practical application.
Response 5: At present, the main restriction on the practical application of alkali-activated slag cementitious materials is its excessive early shrinkage. The use of fly ash and fine sand instead of slag powder can reduce the cost on the one hand, and on the other hand, the activity of both of them is lower than that of slag powder, which may reduce the possibility of early shrinkage of alkali-activated slag cementitious materials, which is also the next work of this paper.
6.Complete the conclusions with the limitations of the proposed methodology. Also write future research..
Response 6: The conclusions have been revised and future studies are described. In the future, the influence of fly ash and fine sand content on the early shrinkage performance of alkali-activated slag cementitious materials will be studied.
7.Generally, the quality of the writing could be improved.
Response 7: Thanks for your comments. The article has been carefully revised from introduction to conclusion.
At last, special appreciation to you for these good questions and comments, and all of them give deep insight and understanding about the results of our work, and greatly improved the quality of our paper.
Thanks again.
Sincerely yours,
Chen-Hui Zhu
Round 2
Reviewer 1 Report
Thanks to the authors for the answers. Unfortunately, they did not display my comments in the new version of the article. For example, the selection of sample sizes for mechanical tests was carried out incorrectly, this should be displayed in the article. The same is for the rest of the comments. The article has hardly changed since the first review.
Author Response
Dear reviewer,
First of all, I'd like to express my apologies. All this is because I did not mark the revision of the article. I have marked up the revisions in the newly uploaded manuscript, and I would like to express my apology again.
The detailed changes are as follows:
(1)The introduction has been revised in the new manuscript, and the disadvantages of existing alkali-activated slag–fly ash cementitious materials such as great shrinkage and easy-cracking are introduced.
(2)The discussion in section 3.1.1 are based on the results of strength test and microscopic test in this experiment, combined with the research results of the research group in the early stage. I have moved this paragraph to discussion (section3.3).
In addition, the selection of sample size for mechanical tests has been displayed in the article. The mold size of the slurry specimen in this test is indeedly 20 mm × 20 mm × 80 mm specimen size. Its flexural strength and compressive strength are tested by DKZ-5000 motorized bending tester and WE-100 hydraulic universal tester (0~100KN).
At last, special appreciation to you for these good questions and comments, and all of them give deep insight and understanding about the results of our work, and greatly improved the quality of our paper.
Thanks again.
Sincerely yours,
Chen-Hui Zhu
Reviewer 2 Report
The paper was improved
Author Response
Dear reviewer,
Thank you for your useful comments and reviewers’ suggestions, and I'd like to apologize for not marking up the changes in the article.
I have marked up the revisions in the newly uploaded manuscript, and I would like to express my apology again.
Thanks again.
Sincerely yours,
Chen-Hui Zhu
Round 3
Reviewer 1 Report
Dear authors,
thank you for your answers. I recommend inserting your answer
"The demolded specimens were placed in water at 20±1℃ for curing. After reaching the curing age, it was removed and wiped on the surface of the specimen, and then covered with a wet cloth until the test. The samples of compressive strength and flexural strength are the same. The samples will first be tested for flexural strength according to the standard (a group of three samples), and the broken samples will be used for the compressive strength test (a group of six samples)."
in section "2.3.2. Test method for mechanical properties ".
Author Response
Dear reviewer,
Thank you for your useful recommendation.
The specific conditions for mechanical testing have been added to the section 2.3.2, and It has also been marked in the manuscript.
Thanks again.
Sincerely yours,
Chen-Hui Zhu